# ECOLE: Learning to call copy number variants on whole exome sequencing data

**Berk Mandiracioglu[1,5], Furkan Ozden[2,5], Gun Kaynar [ ][3], Mehmet Alper Yilmaz[3], Can Alkan [ ][3] & A. Ercument Cicek [ ][3,4] [✉]**

Copy number variants (CNV) are shown to contribute to the etiology of several genetic disorders. Accurate detection of CNVs on whole exome sequencing (WES) data has been a long sought-after goal for use in clinics. This was not possible despite recent improvements in performance because algorithms mostly suffer from low precision and even lower recall on expert-curated gold standard call sets. Here, we present a deep learning-based somatic and germline CNV caller for WES data, named *ECOLE*. Based on a variant of the transformer architecture, the model learns to call CNVs per exon, using high-confidence calls made on matched WGS samples. We further train and fine-tune the model with a small set of expert calls via transfer learning. We show that ECOLE achieves high performance on human expert labelled data for the first time with 68.7% precision and 49.6% recall. This corresponds to precision and recall improvements of 18.7% and 30.8% over the next best-performing methods, respectively. We also show that the same fine-tuning strategy using tumor samples enables ECOLE to detect RT-qPCR-validated variations in bladder cancer samples without the need for a control sample. ECOLE is available at https://github.com/ciceklab/ECOLE.

Copy number variants (CNVs) are well-known and important risk factors for many conditions such as cancer[1,2], schizophrenia[3,4] and autism[5]. High throughput sequencing (HTS) has been the standard technique for the detection of CNVs over the last decade. Various CNV detection algorithms that use whole genome sequencing (WGS) data have been very successful[6–12] with sensitivity and precision values reaching up to 96% and 97%, respectively[13]. This is in contrast to such algorithms working on the whole exome sequencing (WES) data, which suffer from very low precision[14–16]. WGS is a more accommodating platform for this task because it does not employ targeting probes that introduce length, GC, and reference biases[17–19]. On the other hand, WES has been more appealing in the clinic due to being more compact, interpretable, and affordable than WGS. Unfortunately, WES technology has limited clinical use for CNV detection due to these limitations.

We recently developed a deep-learning-based polishing approach which has proven useful in correcting the calls of many state-of-the-art WES-based germline CNV callers using more trustworthy calls made on the matched WGS samples[16]. While this was an important step forward, there are still bottlenecks to making it a feasible option for use in the clinic. The first problem is with the sensitivity of the results. The polisher can only work on the calls (e.g., deletion) returned by the base algorithm. It either changes these calls (e.g., to duplication) or neutralizes them (e.g., to no-call). While this helps to reduce the false discovery rate, it has a limited effect on sensitivity as a polisher cannot make new calls (e.g., convert a no-call to deletion/duplication). Unfortunately, sensitivity has mostly been out of the scope of the WES-based CNV calling domain due to very low performance. The second problem is that even precision performance after polishing is limited on expert-curated CNV call sets which are regarded as the golden ground truth (up to 35%). This is because the polisher uses automated WGS-based CNV calls as labels for model training but these labels (calls) have a very different distribution compared to human expert

[1]Department of Computer and Communication Sciences, EPFL, Lausanne, Switzerland. [2]Department of Computer Science, Oxford University, Oxford, UK. [3]Department of Computer Engineering, Bilkent University, Ankara, Turkey. [4]Computational Biology Department, Carnegie Mellon University, Pittsburgh, PA, US. [5]These authors contributed equally: Berk Mandiracioglu and Furkan Ozden. [✉]e-mail: cicek@cs.bilkent.edu.tr

decisions. Unfortunately, such manually curated call sets are extremely small in size, which prohibits training machine learning models. Thus, a caller that achieves high performance on human expert-curated CNV call sets would enable broad use of WES-based germline CNV detection in the clinic.

Here, we present the first deep learning-based method (ECOLE: Exome-based COpy number variation calling LEarner) which can independently learn to perform somatic and germline CNV calling on WES data. Our model is based on a variant of the *transformer* model[20] which is the state-of-the-art approach to process sequence data in the natural language processing domain[21,22]. ECOLE processes the read-depth signal over each exon. It learns which parts of the signal need to be focused on and in which context (i.e., chromosome) to call a CNV. It uses the high-confidence calls (i.e., labels) obtained on the matched WGS samples as the semi-ground truth. ECOLE improves the exon-wise precision and also the recall of the next best method's performance substantially on a benchmark of automated WGS calls (13.5% and 16.6% improvements, respectively). It is the only method with balanced precision and recall. Moreover, for the first time, we also propose using transfer learning and fine-tuning the model parameters using a small number of human expert-labeled samples. We show that this approach improves the precision and recall by ~18% and ~30%, respectively in predicting human labels. Similarly, we use the fine-tuning method to adapt ECOLE to call somatic variations using bladder cancer samples. We show that we are able to detect PCR-validated copy number aberrations in 13 out of 16 bladder cancer samples while the state-of-the-art method can only detect validated calls in 2 samples even after polishing. With the ability to act as both a germline and a somatic CNV caller and being flexible to adapt to diseases and human experts easily with fine-tuning, we propose ECOLE as a feasible option to broaden the use of exome sequencing technology in the clinic for CNV detection.

## Results

### ECOLE overview

Our model ECOLE is a deep neural network model which uses a variant of the transformer architecture[20] at its core. The transformer is a parallelizable encoder-decoder model that receives an input and applies alternating layers of multi-headed self-attention, multi-layer perceptron (MLP), and layer normalization layers to it. Transformer architecture has achieved state-of-the-art results in signal processing over recurrent neural networks in the natural language processing

domain[20] as well as recently over the convolutional-based models in the computer vision domain[23].

Figure 1 shows an overview of the system architecture. ECOLE takes the read depth over an exon at the base pair resolution. Here, we focus on coding regions only. Thus, the reads mapped outside the exons and the corresponding read depth signal is ignored. This information is transformed into a read-depth embedding using a multi-layered perceptron. We use a classification token to be learned, which is concatenated with the read depth embedding as also done in ref. 23. However, in our setting, this token is chromosome-specific to add further context to the classification task. Finally, the model uses a positional encoding vector which is summed up with the transformed read depth encoding and the classification token. This encoding informs the model on the absolute position of the considered exon. ECOLE applies 3 transformer blocks to this vector. Doing so, it learns the importance of the read depth over a specific base pair, with respect to the read depth on other base pairs, within the same exon region. That is, ECOLE uses an attention mechanism to learn to focus on which base pairs in which context (i.e., deletion, duplication, or no-call). This is in analogy to natural languages where the same word (read depth) having a different stress in different paragraphs (exons) and in different chapters of a text (chromosomes). Finally, we perform classification per-exon using a two-layered perceptron which uses the output of the final transformer block. ECOLE uses higher confidence CNV calls obtained on 1000 Genomes WGS data as the "semi"-ground truth (i.e., compared to WES) to train the model. We use the CNVnator algorithm as the WGS-based germline CNV caller which provides has high sensitivity (86–96%), and high precision (80–97%)[13].

ECOLE is able to transfer the highly accurate decision-making of a WGS-based CNV caller into the WES domain to achieve state-of-the-art performance. Yet, no algorithm in the literature is able to achieve high performance human expert-labeled data is available for a very small number of samples which is insufficient for training a complex model like ECOLE. Here, we apply transfer learning for the first time in the CNV calling domain and create variant ECOLE models tailored towards certain label sets. First, we further tune the parameters of the ECOLE model (trained with the semi-ground truth) using only 4 human expert-labeled samples and generate the ECOLE[FT−EXPERT] model. Second, we fine-tune the parameters of again the base ECOLE model with the MetaSV-based[24] CNV call set generated by the Genome in a Bottle (GiaB) consortium using only the NA12891 sample (Ashkenazi father) and generate the ECOLE[FT−GiaB] model. Finally, to enable the model to

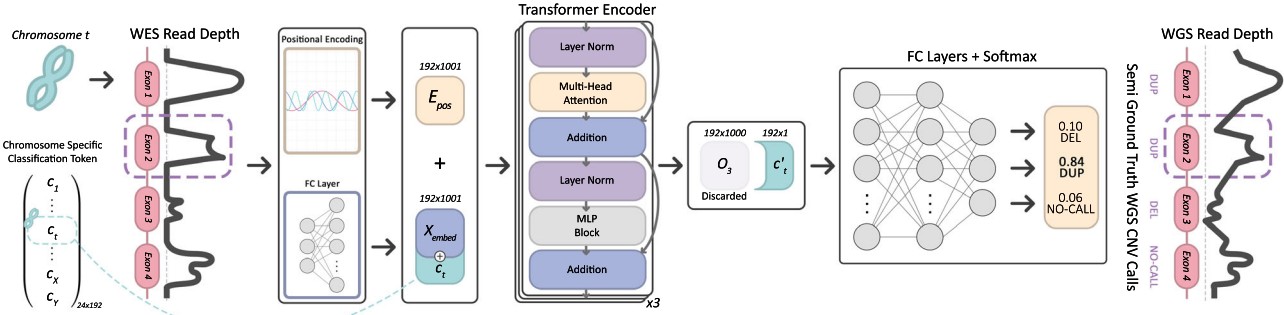

**Fig. 1 | ECOLE's system overview.** The model inputs per exon (i) the read depth signal (length 1000, padded and masked), (ii) chromosome number, and (iii) start and end coordinates of the region. It maps each 1000 read depth value to a higher dimensional vector $X_{embed} \in \mathbb{R}^{192}$ (input embedding) using a fully connected (FC) layer, which is concatenated with a chromosome-specific classification token vector of $c_t \in \mathbb{R}^{192}$ for each chromosome $t$. These chromosome-specific tokens enable the model to learn the chromosome context of the exon samples to perform calls. Transformer layers use a multi-head attention mechanism that learns the connections of each read depth value of base pairs with respect to all other base pairs in the

given exon sample. Therefore, the attention mechanism also learns to which read depth values the classification token needs to pay attention for the respective CNV call. To further learn the positional context of the base pairs within the chromosome, the start and end coordinates of the sample are used to calculate the exon-specific positional encoding $E_{pos} \in \mathbb{R}^{192 \times 1001}$. Two matrices are concatenated and input to a cascade of 3 transformer encoders which generate an output vector $O_3 \in \mathbb{R}^{192 \times 1001}$. Then, the mapped transformation of chromosome-specific classification token $c'_t$ is fetched, which has the size $\mathbb{R}^{192}$. Finally, for the final decision (DEL, DUP, or NO-CALL), we use 2 FC layers followed by softmax activation.

 

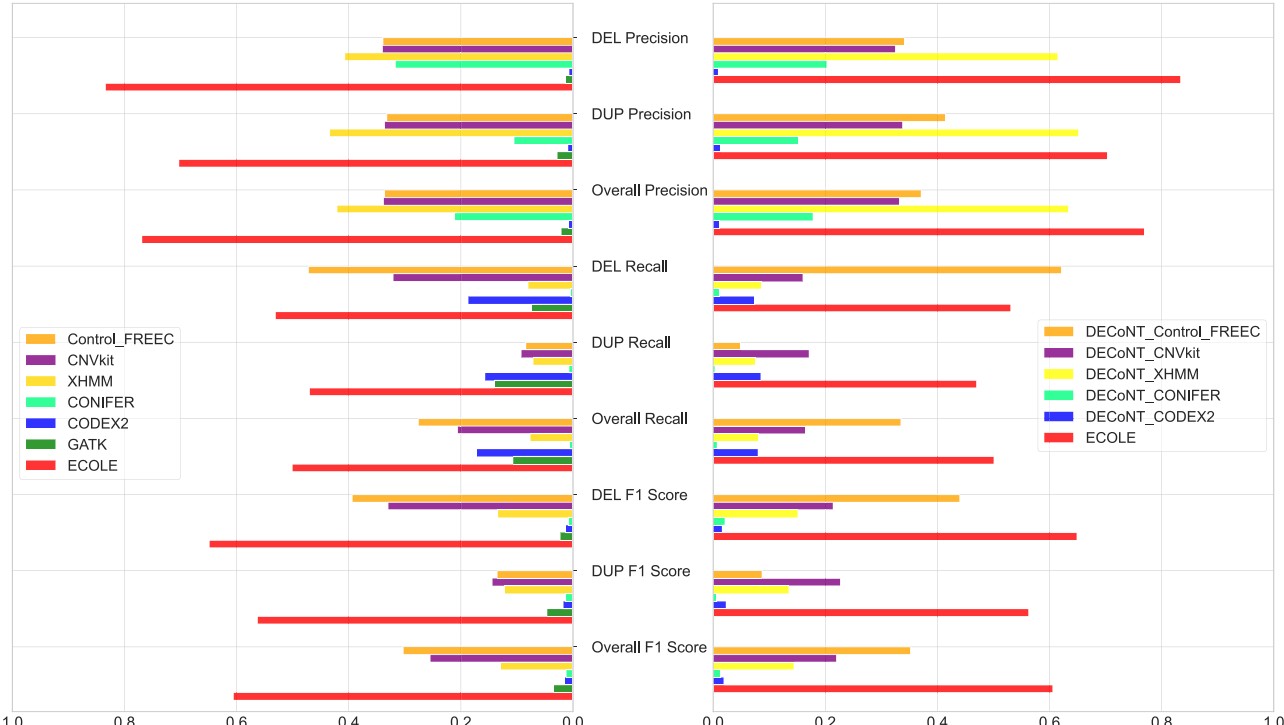

**Fig. 2 | The performance comparison of the WES-based CNV callers on the 1000 Genomes test set which contains 157 samples.** Left: Unpolished results vs. ECOLE and Right: DECoNT-polished results vs. ECOLE. CNVnator calls on the matched WGS samples are used as the ground truth. CNVkit and Control-FREEC return exact (integer) copy number predictions, which are discretized into deletion, duplication, and no-call. We also used the DECoNT tool to polish call sets of all considered tools which are denoted by DECONT-*tool_name*. Source data are provided as a Source Data file.

call somatic CNVs, fine-tune the parameters of the base ECOLE model with bladder cancer samples with semi-ground-truth labels (CNVnator). We call this model ECOLE^FT–SOMATIC.

### ECOLE achieves high performance in WES-based germline CNV calling

**Evaluation criteria.** We consider calls per exon as our fixed evaluation unit. That is, for each exon, ECOLE makes a CNV prediction. For compared methods, we intersect their CNV call segments with the exons, if they report CNVs for larger regions than exons (e.g., merged bins, exons, etc). Each exon has a unique semi-ground truth label (i.e., deletion, duplication, or no-call) assigned with respect to the call made on WGS data of the same sample. See Supplementary Fig. 1 for the visual demonstration of this procedure.

**CNV calling performance of ECOLE on a WGS-based semi-ground truth call set.** We compare the performance of ECOLE with the state-of-the-art germline CNV callers from the literature on the 1000 Genomes WES samples (test split, see Section 4.1 for data set details). The semi-ground truth CNV calls are obtained using CNVnator on the WGS samples of the same individuals. We show the distribution of deletion and duplication call sizes in the training set in Supplementary Figs. 2 and 3. Compared methods are XHMM, CODEX2, CONIFER, Control-FREEC, GATK[17,25–29]. Among those, CNV-kit and Control-FREEC predict integer copy numbers while the others report the CNV (i.e., deletion or duplication). To be able to fairly compare the performance with them, we discretize their predictions. We also polish the call sets of these tools using available call-polisher DECoNT models and compare ECOLE with the polished versions of the call sets of these algorithms (See Section 4.2 for details of compared methods).

Figure 2 shows the precision, recall, and F1 score results for each algorithm. Supplementary Table 1 shows the corresponding values and Supplementary Table 2 shows the respective confusion matrices.

ECOLE achieves the best average precision values over even polished versions of the other algorithms and provides 13.5% improvement over the next best performance by DECoNT polished XHMM call set (DECoNT-XHMM). Also in terms of deletion and duplication precision, we provide the best results with 21.9% and 5.2% improvements, respectively. ECOLE achieves 50.1% overall recall which is a 16.6% improvement over the second-best model, DECoNT-Control-FREEC. While ECOLE is able to achieve high recall and it is also the first method that is able to balance precision and recall. ECOLE yields an F1-score of 60.6% which corresponds to an improvement of 25.4% over the second-best result obtained by the DECoNT-Control-FREEC call set For all other methods, if the precision is high, the recall is low due to the small number of calls made and if the recall is high, the precision is low due to the large number of predictions made. Please also see the precision-recall curve of ECOLE in Supplementary Fig. 4. We also analyzed the specificity (NPA) and the negative predictive value (NPV) performance of ECOLE and compared it with other tools. We observe that ECOLE achieves 99.9% overall NPA and outperforms other tools with 99.6% overall NPV. Please see Supplementary Tables 3 and 4 for the detailed NPV and NPA results, respectively.

We also compare ECOLE with CNLearn which is a random forest-based method that creates an ensemble of four WES-based callers (See Section 4.2 for details). We compare our results on the 28 samples for which we obtained results via personal communication with Santhosh Girirajan. As shown in Table 1 ECOLE performs substantially better in all metrics considered, and see Supplementary Table 5 for the corresponding confusion matrix.

### CNV calling performance generalizes to other sequencing platforms and capture kits.

The WES data we used to train the ECOLE model were obtained using Illumina HiSeq 2000 and Illumina Genome Analyzer II platforms. Here, we show that ECOLE's performance generalizes to other sequencing platforms that are not used during

**Table 1 | Performance comparison of ECOLE with CNLearn on 28 samples from the 1000 Genomes Project**

| Tool | DEL Precision | DUP Precision | Overall Precision | DEL Recall | DUP Recall | Overall Recall | DEL F1 Score | DUP F1 Score | Overall F1 Score |
|---|---|---|---|---|---|---|---|---|---|
| CNLearn | 0.084 | 0.221 | 0.152 | 0.002 | 0.010 | 0.006 | 0.004 | 0.019 | 0.012 |
| ECOLE | 0.834 | 0.679 | 0.757 | 0.541 | 0.500 | 0.520 | 0.656 | 0.675 | 0.617 |

Note that these 28 samples are not included in the training set of ECOLE and the predictions are obtained via personal communication with the authors.

training. Here, we test the ECOLE model using the sequencing data of the NA12828 sample obtained using (i) BGISEQ 500, (ii) HiSeq 4000, (iii) NovaSeq 6000, and (iv) MGISEQ 2000. We did not use any related data for this sample during the training process.

The results are shown in Supplementary Table 6 and Supplementary Fig. 5. See Supplementary Tables 7–10 for the corresponding confusion matrices. We observe that ECOLE is the best-performing method in all categories with overall F1-scores ranging between 49.9% and 58.6%. Note that the performance for BGISEQ and MGISEQ platforms is relatively more important for this set of experiments as these platforms are built by an entirely different manufacturer. In BGISEQ and MGISEQ, we observe that the ECOLE remains to be the best-performing tool with respect to all considered benchmarks, providing at least ~14% overall F1-score improvements over the second-best method, DECoNT-Control-FREEC. Once again, ECOLE is the only method with balanced precision and recall. Similarly, in NovaSeq 6000 and HiSeq 4000 platforms, we observe ~40% and ~30% overall F1 score improvements.

These results demonstrate the robustness of our model in dealing with systematic biases and noise introduced by different systems. We show that our model can be used across platforms when there is not enough WGS-matched data samples to train a ECOLE model obtained on the platform of interest.

We also analyze the effect of the WES capture assay design on ECOLE's CNV calls. We compare ECOLE's performances on the NA12878 samples sequenced with NimbleGen SeqCap v3 and SeqCap EZ Human Exome Library v3.0 capture kits which cover 99.3% and 67.8% of the exome by single probes, respectively. About 85% of probesets are overlapping with each other. As it can be seen in Supplementary Table 11, ECOLE achieves similar scores for both capture kits. It's expected because even if the breakpoint does not fall into the same capture region within the same exon, the model is informed about the read depth and the label information of other exons within the same chromosome through the positional encoding and the chromosome specific classification token. So, read depth differences and decisions made for other exons affect the decisions made for the exons.

We observe that probe count does not substantially affect the model performance for the SeqCap EZ Human Exome Library v3 capture kit (see Supplementary Table 12). For NimbleGen SeqCap v3, performance on single probe-covered exons is better compared to multiple probe-covered exons. There are only a few multiple probe-covered exons (0.7%) by NimbleGen SeqCap v3 and they mostly reside in chromosomes 9 and 10 (51%). As discussed in Section 2.5, ECOLE performs relatively worse even when predicting on samples sequenced with the same capture kit. This might explain the reason for the low performance in multiple probe-covered exons in NimbleGen SeqCap v3. We also stratify exons with respect to their GC contents and compare ECOLE's performance on varying GC content rates for both capture kits. We observe that GC content does not substantially affect the overall performance. The results can be seen in Supplementary Figs. 6 and 7.

**CNV calling performance on human expert calls.** Here, we use the highly validated CNV call set produced by Chaisson et al.[30] as the ground truth to test the performance of the WES-based CNV callers. Note that this call set contains CNV calls for 9 individuals from the 1000 Genomes Project WGS samples. This is a human expert-curated,

consensus call set that relies on the results of 15 WGS-based CNV callers compared against structural variations generated using PacBio with single base pair breakpoint resolution. We use 8 samples from this call set that have matching WES data. Calls on 4 of these samples are used for training and the rest are used for testing (see Methods for details). We show the distribution of deletion and duplication event sizes of the test set (Chaisson et al.) in Supplementary Figs. 8 and 9.

Results are shown in Fig. 3. Please see Supplementary Table 13 for the values in this figure and Supplementary Table 14 for the corresponding confusion matrices. All compared CNV callers and their polished versions have much lower F1-score performance on predicting human expert calls compared to predicting the WGS-based semi-ground truth labels (i.e., CNVnator calls). The top F1-score performance reaches up to ~10% as opposed to ~20%, and no algorithm shows balanced precision and recall. These are in line with the observations in ref. 16.

We also observe that ECOLE exhibits low performance. It provides only 3.7% overall F1 score improvement over CONIFER which is the next best method. This is expected as the label distribution on this data set is different from what ECOLE is trained with. Samples from Chaisson et al. study have 4 times more DEL calls and 2 times fewer DUP calls compared to samples in our training set which are labeled by CNVnator. This call set is more than two orders of magnitude smaller than what we use to train ECOLE which prohibits training an ECOLE model from scratch.

To address this issue, we use transfer learning and use the left-out 4 samples from Chaisson et al. to fine-tune the parameters of the trained ECOLE model. That is, we further train the final ECOLE model using the human expert labeled samples and adjust the model weights. We call this fine-tuned model ECOLE[FT−EXPERT]. Note that none of the other methods have a way of incorporating this information.

We observe that ECOLE[FT−EXPERT] outperforms all other methods including the baseline ECOLE with an overall precision of 68.7% and an overall recall of 49.6%. It effectively balances precision and recall and obtains the top F1-score in all categories. It provides substantial improvements in F1-scores with 42.6%, 50.5%, and 46.8% increases over the next best method in deletion, duplication, and overall F1-scores, respectively. ECOLE[FT−EXPERT] also achieves better NPV and NPA than ECOLE with 99.4% and 99.7% overall scores, respectively. These results show that ECOLE[FT−EXPERT] have more accurate positive and negative predictions than ECOLE. ECOLE[FT−EXPERT] reduces the number of false negatives for duplication, and deletion calls by 1088 and 5234, respectively. Please see Supplementary Tables 15 and 16 for detailed NPV and NPA results, respectively.

To test if fine-tuning works on an independent call set, we fine-tune the base ECOLE model with the call set provided by GiaB for the Ashkenazi father (NA12891) to obtain the ECOLE[FT−GiaB] model. We test this model on the Ashkenazi mother (NA12892). The base ECOLE model achieves 0.8% precision and 8.1% recall and ECOLE[FT−EXPERT] achieves 1.25% precision and 5.5% recall for this sample. On the other hand, ECOLE[FT−GiaB] achieves 68.6% precision and 58.6% recall. This result shows that even a single sample with labels is effective in configuring the model to work on an independent CNV call set.

**Somatic CNV calling performance.** ECOLE is a germline CNV caller by design as it is trained with normal tissue samples. Similar to the difference between the automated WGS-based calls and the human

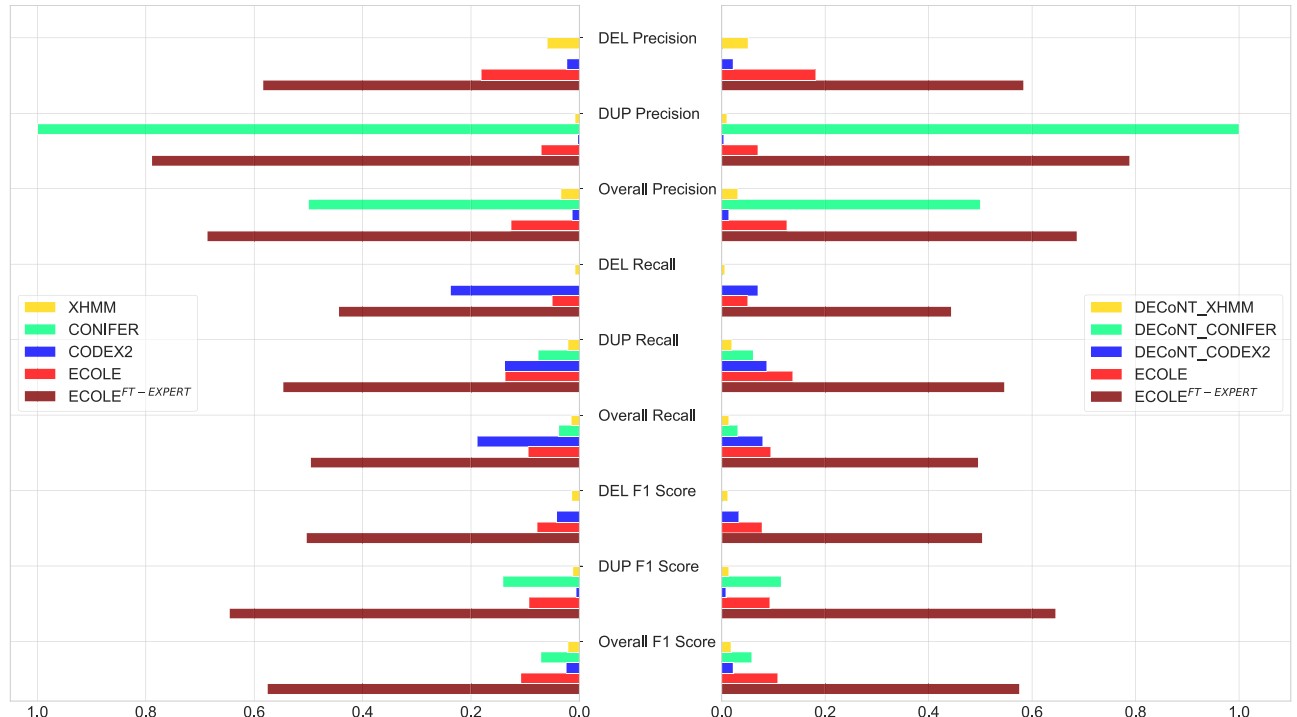

**Fig. 3 | The performance comparison of the WES-based CNV callers on the 1000 Genomes test set which contains 157 samples.** Chaisson et al.'s human expert-curated calls on the matched WGS samples are used as the ground truth[30]. We also used the DECoNT tool to polish call sets of all considered tools which are denoted by DECONT-*tool_name*. ECOLE[FT-EXPERT] corresponds to the fine-tuned version of ECOLE model with human expert calls. Source data are provided as a Source Data file.

expert calls, germline CNV calls and somatic CNV calls have different distributions. This is due to the difference between the WES read depth signal of the tumor and control samples. For this reason, specific callers or specific modes of callers are designed for somatic CNV calling which often requires paired control and tumor samples to account for the difference which increases the computing and sequencing cost.

Here, using the same fine-tuning strategy, we update the parameters of the ECOLE model with bladder cancer samples from[31] (SRA: SRP017787). This study reports matched WES and WGS samples of 16 bladder cancer samples and RT-qPCR validated CNVs in 4 regions. These events coincide with the following genes and affect the corresponding samples: A deletion in *CDKN2A/B* (samples B63, B112, and B80-0), a duplication in *CCDN1* (samples B37 and B103), a duplication in *DHFR* (samples B15, B18, B19, B24, B34, and B50) and a duplication in *ERBB2* (samples B9, B23, B80, B80-5, and B86) genes.

We fine-tune ECOLE to ECOLE[FT-SOMATIC] using (i) the CNVnator semi-ground truth labels obtained on the WGS data of samples B112, B24, and B80 and (ii) the corresponding WES read depth signal obtained on the matched WES data of samples B112, B24, B80. We use the remaining 13 bladder cancer samples to test if we can detect the RT-qPCR-validated CNVs for each sample. We compare ECOLE[FT-SOMATIC] with XHMM which consistently obtains the highest precision, its polished call set DECoNT-XHMM, and ECOLE.

As shown in Table 2, XHMM is able to detect the validated deletion event in the *CDKN2A/B* gene for one sample (B112) and does not return any calls for the remaining 10 samples. The polished version of XHMM's call set verifies these calls. ECOLE does not make any calls for any of the samples in the validated regions. On the other hand, ECOLE[FT-SOMATIC] is able to detect all of the 13 validated CNVs in the corresponding 13 test samples (all samples except the samples used in the fine-tuning). This shows that the model is flexible and can be easily configured to make somatic calls even without the need for a control sample.

We also computed the genome-wide precision, recall, and F1-score performances with respect to the semi-ground truth labels obtained on the matched WGS data of the 13 test samples obtained using CNVnator. Please refer to Supplementary Table 17 for the corresponding confusion matrices. We find that ECOLE has both lower precision and lower recall than others. Table 3 and Supplementary Fig. 10 show that ECOLE[FT-SOMATIC] outperforms others and provides an F1 score improvement of 25.2% over the next best method which shows that fine-tuning improves the performance (See Supplementary Table 17 for the corresponding confusion matrix). ECOLE[FT-SOMATIC] trades some precision of ECOLE for a large gain in the recall. We wanted to make sure that fine-tuning does not act as a simple threshold that is relaxed so that ECOLE[FT-SOMATIC] makes more calls than ECOLE to achieve higher recall. For this, we relaxed the call threshold of ECOLE to make it more liberal (i.e., it makes a call even if the probability is less than 0.33). Despite the increase in recall in this case, ECOLE was not able to a make call for any of the validated regions. This shows us that fine-tuning effectively teaches the algorithm about making calls in somatic samples and does not serve as a simple filtering mechanism.

## CNV calling performance on merged CNV segments
**Evaluation Criteria.** WES-based CNV callers often make calls for exons or bins which sometimes exceed exon bounds and then use a segmentation method to merge the subsequent calls into a larger call region.

On the other hand, the ground truth calls on the WGS data are often shorter. A merged call on the exome can span multiple WGS-based calls. To assign a WGS-based semi-ground truth label to the WES-based call, the covered calls made on the WGS data are merged and a consensus label is assigned[16]. Supplementary Fig. 11 exhibits this procedure visually for further reference.

This procedure comes with the following problems: First, it reduces the resolution in the ground truth due to smoothing. Second,

**Table 2 | CNV calls for the RT-qPCR validated regions of 16 bladder cancer samples from Guo et al.[31]**

| Gene | Chromosome | Region Start | Region End | Call | Sample Name | XHMM | DECoNT-XHMM | ECOLE | ECOLE[FT-SOMATIC] |
|---|---|---|---|---|---|---|---|---|---|
| CDKN2A/B | 9 | 20.3 m | 24,1 m | DEL | B63_Cancer | No | No | No | **Yes** |
| CDKN2A/B | 9 | 20.3 m | 24,1 m | DEL | B112_Cancer | **Yes** | **Yes** | No | N/A |
| CDKN2A/B | 9 | 20.3 m | 24,1 m | DEL | B80-0_Cancer | No | No | No | **Yes** |
| CCDN1 | 11 | 69.8 m | 69.8 m | DUP | B37_Cancer | No | No | No | **Yes** |
| CCDN1 | 1 | 69.8 m | 69.8 m | DUP | B103_Cancer | No | No | No | **Yes** |
| DHFR | 5 | 79.9 m | 80 m | DUP | B15_Cancer | No | No | No | **Yes** |
| DHFR | 5 | 79.9 m | 80 m | DUP | B18_Cancer | No | No | No | **Yes** |
| DHFR | 5 | 79.9 m | 80 m | DUP | B19_Cancer | No | No | No | **Yes** |
| DHFR | 5 | 79.9 m | 80 m | DUP | B24_Cancer | No | No | No | N/A |
| DHFR | 5 | 79.9 m | 80 m | DUP | B34_Cancer | No | No | No | **Yes** |
| DHFR | 5 | 79.9 m | 80 m | DUP | B50_Cancer | No | No | No | **Yes** |
| ERBB2 | 17 | 35 m | 35.2 m | DUP | B9_Cancer | No | No | No | **Yes** |
| ERBB2 | 17 | 35 m | 35.2 m | DUP | B23_Cancer | No | No | No | **Yes** |
| ERBB2 | 17 | 35 m | 35.2 m | DUP | B80_Cancer | No | No | No | N/A |
| ERBB2 | 17 | 35 m | 35.2 m | DUP | B80-5_Cancer | No | No | No | **Yes** |
| ERBB2 | 17 | 35 m | 35.2 m | DUP | B86_Cancer | No | No | No | **Yes** |

The table lists the genes, regions, validated calls, and the predictions of each method. Note that ECOLE[FT-SOMATIC] is fine-tuned on samples B112, B24, and B80. The calls of ECOLE[FT-SOMATIC] for these samples are denoted as N/A as they are used during training. Bold denotes captured CNV.

**Table 3 | Somatic CNV calling performance comparison on 13 bladder cancer test samples from Guo et al.[31]**

| Tool | DEL Precision | DUP Precision | Overall Precision | DEL Recall | DUP Recall | Overall Recall | DEL F1 Score | DUP F1 Score | Overall F1 Score |
|---|---|---|---|---|---|---|---|---|---|
| XHMM | 0.235 | 0.962 | 0.698 | 0.012 | 0.028 | 0.020 | 0.023 | 0.054 | 0.038 |
| DECoNT-XHMM | 0.193 | 0.950 | 0.572 | 0.010 | 0.023 | 0.017 | 0.019 | 0.045 | 0.033 |
| ECOLE | 0.373 | 0.673 | 0.523 | 0.019 | 0.010 | 0.015 | 0.036 | 0.020 | 0.029 |
| *ECOLE*[FT-SOMATIC] | 0.243 | 0.423 | 0.333 | 0.147 | 0.372 | 0.260 | 0.183 | 0.395 | 0.292 |

CNVnator calls are used as the semi-ground truth to calculate the metrics.

this results in the ground truth changing with respect to the breakpoints of calls made by each WES-based caller. This makes it impossible to form a global ground truth call set to calculate recall. It was not a problem earlier in the literature as methods were mostly focused on precision. Here, we compare the precision of ECOLE with others when we merge the exon-level calls to obtain larger call segments that also cover noncoding regions. Note that, ECOLE works at a base-pair resolution and makes a call per exon. Here, we merge subsequent exons with the same call to obtain a merged CNV segment to compare with other algorithms which often rely on a segmentation step, and compare the precision performance.

Supplementary Table 18 and Supplementary Fig. 12 show the precision of each algorithm for the samples in the 1000 Genomes Dataset test split. We use the merged CNV segments as predictions for all algorithms and use merged the semi-ground truth labels obtained on the WGS data for the same samples. We can observe that ECOLE is able to perform comparably to the top performing tool (DECoNT-XHMM) in terms of precision. This is still important as ECOLE achieves this precision quality while maintaining over 18% improvement in the average recall metric. Evidently, ECOLE is able to make calls on a greater scale (merged CNV segments) just as it is able to perform on high resolution (i.e., exon-level).

We show the size distribution of deletion and duplication calls (merged CNV segments) in the training set in Supplementary Figs. 13 and 14. We stratify the deletion and duplication performances of ECOLE with respect to the call sizes. As shown in Supplementary Figs. 15 and 16, ECOLE performs well for a wide range of exon sizes. Specifically, ECOLE's performance for deletion calls is low for small exons (50–100 bp) and exons that are longer than 4,000 bp. This is

mainly due to the small number of samples in these size ranges. For the former, it is also due to the signal being quite short and thus possibly being noisy for the model to generalize. As for the duplication performance, we do not have any events at the 50–100 bp range (excluded from the figure) and see a similar decrease of recall for very long exons.

The comparison of precision performances with CNLearn is provided in Supplementary Table 20. We observe that ECOLE has better precision than CNLearn. It provides a 49.3% improvement on average precision while providing a substantial average recall improvement as discussed before.

Supplementary Table 22 and Supplementary Fig. 17 show the precision performances of every method when using the human expert-curated labels as the ground truth[30]. See Supplementary Table 23 for the corresponding confusion matrices. We obtain a 14.3% average precision improvement over the next best method, CONIFER. While CONIFER achieves perfect precision in the DUP category, it has zero precision in the DEL category and it only makes a handful of calls. The actual second best-performing method with an acceptable number of calls is polished CODEX2 which is 30% behind ECOLE[FT-EXPERT].

Supplementary Table 24 and Supplementary Fig. 18 shows the performances of the tools on the NA12878 sample which was sequenced on various platforms. ECOLE is able to maintain its pre-eminence over all performance metrics when merged CNV segments are considered. We observe that our model is providing at least ~28% average precision improvement over the second-best performing method in all the sequencing platforms considered.

Finally, we investigate the performance of ECOLE on single exon events, which are critical to detect in clinical use cases. We see that ECOLE has 56.9% overall precision and 78.7% overall recall for single

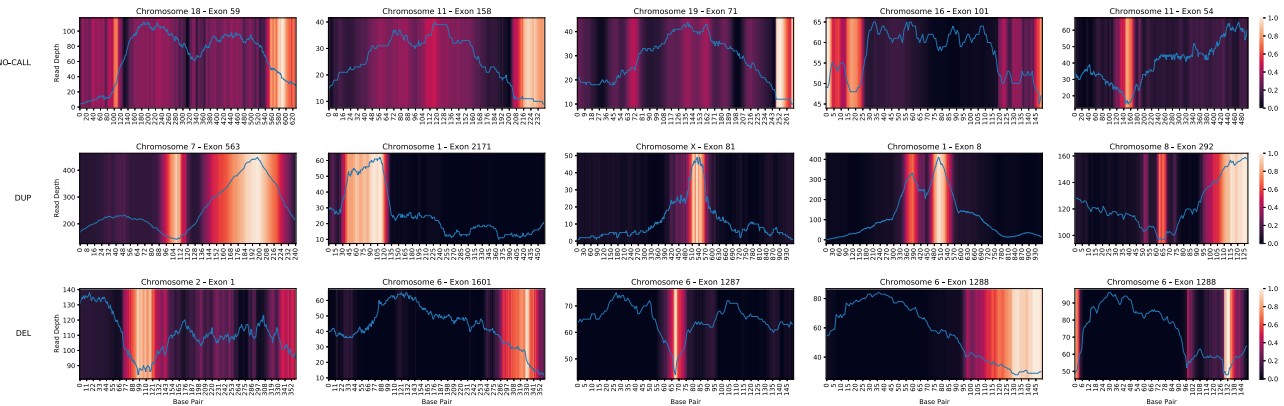

**Fig. 4 | The figure shows the read depth signal over 15 exons with 5 NO-CALLs (row 1), 5 DUP calls (row 2), and 5 DEL calls (row 3).** Heatmaps in the background denote the relevance score of the corresponding part of the signal assigned by the model. The brighter the color the higher the attention devoted to that region. For each panel, the x-axis denotes the index of the base pair, the left y-axis denotes the read depth value, and the right y-axis denotes the relevance score (attention). Source data are provided as a Source Data file.

exon events. Please see Supplementary Figs. 19 and 20 for the distribution of the number of exons in deletion and duplication calls in the merged CNV segments. We stratify ECOLE's deletion and duplication performance by the number of exons in merged segments in Supplementary Figs. 21 and 22. We find that ECOLE's deletion detection performance on single-exon events is lower (~50% F1-score) than on multi-exon events (~70% F1-score). We observe an increase in ECOLE's deletion detection performance as the number of exons in the call increases. On the other hand, ECOLE's duplication detection performance on single-exon events is on par with the multi-exon event performance (~80% F1-score).

We conclude that ECOLE improves the state-of-the-art CNV calling precision even when outputting merged CNV calls instead of exon-level calls. Note that this is a disadvantageous benchmark setting for our approach as our approach works on a base-pair resolution and the merge process decreases the resolution of our calls.

**Interpretability of the CNV calls**

Transformer-based neural networks are inherently interpretable as they incorporate an attention mechanism. The attention component of the network learns which parts of the read depth signal have to be focused on by the model to make the decision, similar to humans selectively focusing on certain parts of an image to recognize. However, it is not straightforward to visualize the parts of the read depth signal focused by ECOLE since the model uses a multi-head attention mechanism which means multiple attentions are calculated over the signal which is then concatenated and transformed (linear) into the same dimensions as input (192 x 1001). Therefore, there is an implicitly learned complex relationship between these attention maps that the model uses to get the final decision. As Voita et al. demonstrate that every attention head carries different importance for the final classification and a simple average over the multiple heads causes noisy relevance maps for visualization[32].

We use the Generic Attention-model Explainability method proposed by Chefer et al. to visualize the parts of the signal that are deemed important for making the CNV calls[33]. Figure 4 shows the read depth signal observed over 15 exons. The background heatmap indicates which parts of the signal are attended by the model where brighter color indicates more attention. ECOLE classifies the examples in the first row as NO-CALL, the second row as duplication, and the last row as deletion. For the duplication calls, the sharp shifts in read depth signals, mostly elevations, were focused by the model. Likewise, for deletion calls, we can observe that the model focused on locations that have sharp downfalls of read depth values. For both cases, the rest of the signal receives almost no attention and is ignored by the model.

For the exons with no calls, we observe that the model still focuses on the inclines and declines in the read depth signal, but other parts of the signal receive relatively more attention compared to the exons with calls. Since the model cannot detect a concrete pattern and is not confident enough, it opts for a no-call.

This is a nice feature of ECOLE as the user (e.g., the clinician) can visualize the reasoning behind ECOLE making a CNV call over an exon and check if the change in the read depth signal is credible to make a call. The beginning and the end of the attention ranges implicitly might correspond to the breakpoints as the method learns that regions with such sharp changes are important to make calls but this does not have to be the case. This is because the model also takes into account the context, which is the read depth across other exons in that chromosome to make a decision.

**Insights from ECOLE's CNV calls**

First, we focus on ECOLE's calls made on pseudoautosomal regions of Chromosome X - PAR1 and PAR2 which are diploid regions and are usually problematic for CNV callers. We compare the performance with XHMM. The polished XHMM call set has a precision of 37% and 50% in these regions, respectively. On the other hand, ECOLE achieves a precision of 73.6% and 73.8%, respectively. On the X chromosome as a whole, ECOLE has an exon-wise precision of 65% whereas polished XHMM has a precision of 16%. We find that the model is very conservative in making DEL calls for the males on the X chromosome. It makes 6 DEL calls and attains a precision of 50%. While the performance is not perfect, the result shows that the model is not making spurious deletion calls for males due to lower read depth signal and learns to correct this issue as expected. We also analyze the performance of ECOLE on the segmental duplication results to see how our models behave in hard-to-map regions. We use segmental duplication region data from UCSC dataset (https://genome.ucsc.edu/cgi-bin/hgTables?clade=mammal&org=Human&db=hg38&hgta_group=allTracks&hgta_track=genomicSuperDups&hgta_regionType=genome). ECOLE achieves a precision of 87.1% for the 157 test samples from the 1000 Genomes dataset. These results show that ECOLE performs well in this challenging setting.

Figure 5 shows the chromosome-wise stratification of the calls where each dot represents a call made by ECOLE for each sample on our test set (1000 Genomes WES data): Green if the call is correct with respect to matching WGS-based semi ground-truth calls and gray otherwise. We observe that the method's performance is lower for very short exons (less than 100 bps). This is expected as the read depth signal length in these regions is shorter and is more prone to noise as the method is input with less information. The length distribution of

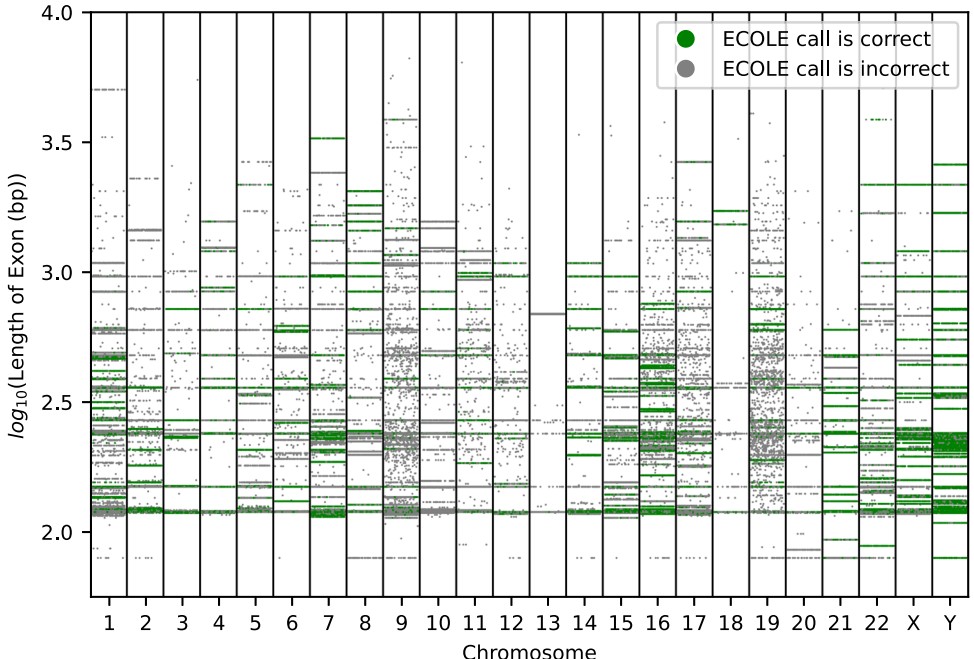

**Fig. 5 | Each dot in this figure represents a CNV call by ECOLE for each sample on our test set (1000 Genomes WES data).** The ground truth is the CNV calls made by CNVnator on the corresponding WGS samples. We separated each call based on the chromosomes. The y-axis of the plot shows the length of exons (bp) in the log scale. Green dots represent a correct CNV call made by ECOLE for that sample, whereas grey dots represent an incorrect call made by ECOLE. We added jitter to each sample on the x-axis for better visualization. Source data are provided as a Source Data file.

exons is shown in Supplementary Fig. 23 and there are only very few exons with length less than 100bp which makes it harder for the model to generalize We also observe that the success of the method varies across chromosomes. The method performs well in chromosomes 14, 21, and Y with accuracy reaching up to 80%. On the other hand, the performance is lower on chromosomes 9, 10, and 13, where the accuracy is below 10%. Except for chromosome 9, these are chromosomes with short exons and a relatively low number of calls which might explain why the model had difficulty in learning the true distribution of the calls.

Finally, we analyze the effect of read depth changes on the performance of the model. We sub-sample the reads for the NA12892 sample (originally 200x coverage) at varying rates (80%, 60%, and 40%) and compare the performance of ECOLE[FT−GiaB] with decreasing coverage on this sample. As shown in Supplementary Fig. 24, precision and recall tend to decrease and sometimes stay constant with some noise. The lowest F1-score is close to 40% even at 80x coverage. We think the results show that the method can perform reasonably well even at very low coverage data but we suggest higher coverage for better performance.

## Discussion

Copy number variants have a large spectrum of phenotypic effects from just playing a role in genetic diversity to underlying complex genetic disorders by affecting roughly 10% of the genome[34]. Accurate CNV calling on WES data for use in clinics has been a long sought-after goal due to cost, size, and time advantages compared to WGS. Indeed with its high diagnostic yield, WES has been a mainstream tool in routine practice in genomic medicine[35]. Yet, WES-based CNV callers have suffered from low precision and concordance[14,15]. As we have recently shown, it has been possible to transfer the satisfactory CNV calling performance of WGS-based CNV callers to the WES-based callers, using a deep learning-based polishing approach[16]. Polishing selectively prunes out false positive CNV calls and substantially improves precision. However, by design, a polisher cannot make new

calls as it is dependent on the calls of the base caller. While it is possible to change a false positive to a true negative call, it is rare and it is not possible to change a false negative call (i.e., no call) to a true positive call. This hinders improving the recall. Here, we show that it is possible to use deep learning techniques to process the read depth signal and train a stand-alone WES-based CNV caller which is able to achieve WGS-level precision and recall performance at the same time.

We use WGS-based calls as labels to train our model obtained using CNVnator. These must be regarded as semi-ground truth rather than absolute ground truth data as CNVnator reports an 86-96% recall and a 3-20% false discovery rate. The ideal case is using a human expert-curated set of calls to train ECOLE. However, such a call set is only available for 9 samples from the 1000 Genomes Dataset[30]. Unfortunately, it is orders of magnitude smaller compared to the CNVnator call set and it is not possible to be able to train a complex model like ECOLE. As human expert decision-making does not resemble the decision-making of automated tools, the overall precision in predicting human expert calls even after polishing was limited at 35%[16]. Here, we show that it is possible to use a pretrained ECOLE model and further train it using this limited set of human calls. This is called fine-tuning in machine learning literature. That is, we take the model trained with large-but-not-fully-confident WGS-based calls and then continue training with small-but-confident human expert calls. We show that fine-tuned ECOLE (ECOLE[FT−EXPERT]) is the first method to achieve high performance on this call set. Similarly, germline CNV calling and somatic CNV calling differ due to the difference in typical read depth signatures between a control and a tumor sample. We use the fine-tuning strategy to convert ECOLE, which is a germline CNV caller, into a somatic CNV caller using matched WES and WGS tumor samples. ECOLE[FT−SOMATIC] is specific to bladder cancer as the samples we used were as such. However, the storage, computational and time cost of configuring ECOLE into any cancer type of interest is very low as the model requires only a few samples and only a few epochs for the model update. We think with high performance on human expert call set and the ability to perform accurate somatic CNV calling, ECOLE is a

promising candidate CNV caller to use in the clinic. If one would like to use an ECOLE model with a different CNV calling strategy than available ECOLE models, they can label CNVs on a few WES data with their technique and fine-tune the base ECOLE model. This would tweak the decision-making of ECOLE towards their strategy. The tool is easily configurable to other specific CNV calling techniques without generating WES data.

We observe some limitations and potential future directions. For instance, very short exons are more difficult for ECOLE to generalize and the performance is relatively stable across exon sizes otherwise (Supplementary Figs. 15 and 16). The performance is relatively lower for chromosomes with a small number of examples. While ECOLE is released as an organism or condition-agnostic tool for broad use, it is possible to incorporate prior condition-specific knowledge into the model to make it work in a more optimized fashion for such regions or conditions. For instance one can group certain nearby exons to have a longer and more informative read depth signal or one could let the chromosome-specific tokens be shared across some chromosomes to increase the performance in relatively low-performing chromosomes. The model is designed to work with the read depth signal only but using more information from the aligner would potentially benefit the calling performance and may enable detecting balanced rearrangements such as flagging regions with split reads, discordant read pairs or reads with aberrant mapping orientation.

## Methods
### Data Sets
**Training and test sets from the samples in the 1000 Genomes Project.** We use 707 samples from the 1000 Genomes Project[36] to train and test ECOLE. This corresponds to samples HG00096 to HG02356 in alphabetical order. We use both the WES and the WGS data for each sample. The WES data were sequenced with Illumina Genome Analyzer II and Illumina HiSeq 2000 while the WGS data were generated using NovaSeq[36] and for WES data NimbleGen SeqCap v3 capture kit was used. The average read depth is 50× for WES and 30× for WGS data with average read lengths of 76 bps and 100 bps, respectively. BWA-MEM is used for alignment on GRCh38[37]. We use the CNVnator[13] tool to call CNVs on the WGS data of each sample to obtain the semi-ground truth labels. The training and test sets consist of 550 and 157 samples, respectively. We use the training set to train and obtain the final ECOLE model where the WES read depth is used as the input and the WGS-based CNVnator labels are used as the semi-ground truth. The training samples are labeled with 740,178 DEL and 953,202 DUP calls in total. The true negatives are the rest of the exons for which no calls are made. The test set is used to evaluate the performance as shown in Figure 2 and Table 1. Supplementary Dataset 1 lists the corresponding names of the samples. The test samples are labeled with 210,597 DEL and 282,698 DUP calls in total.

**NA12878 samples for generalizability tests.** We use the calls made for the NA12878 sample to test the generalizability of ECOLE to various sequencing platforms. This sample has WES data provided by the following platforms: BGISEQ 500, Illumina HiSeq 4000, MGISEQ 2000, and NovaSeq 6000. We use this sample only for testing and its data is not included in the training set by any means. Again, we use the CNVnator calls on the WGS sample of NA12878 to obtain the semi-ground truth labels per exon for the evaluation. This sample is labeled with 1,780 DEL and 1,350 DUP calls in total.

**Fine-tuning and test sets from the samples in Chaisson et al. call set.** Chaisson et al.[30] provide human expert-validated consensus calls of 15 CNV WGS callers on 9 samples from the 1000 Genomes project. We obtained the calls made for the 8 samples, for which there is also matching WES data in the 1000 Genomes dataset, namely: HG00512, HG00513, HG00731, HG00732, HG00733, NA19238,

NA19239, NA19240. The calls made by Chaisson et al. on the WGS data were used as the golden standard ground truth for all compared algorithms and ECOLE. We used the ground truth calls made for 4 samples (NA19238, NA19239, HG00731, HG00512) to fine-tune the parameters of ECOLE when applying transfer learning. These samples are labeled with 16,445 DEL and 2,566 DUP calls in total. The true negatives are the rest of the exons for which no calls are made. We used the remaining 4 samples (HG00513, HG00732, HG00733, NA19240) for the test (inference) and comparison with other tools. The test samples are labeled with 14,496 DEL and 2,624 DUP calls in total. See Supplementary Dataset 1 for the corresponding names of the samples.

**Fine tuning and test sets from the samples in GiaB.** GiaB provides the MetaSV-based CNV call set for the Ashkenazi family (NA12878 - son, NA12891 - father and NA12892 - mother). We fine-tune the ECOLE model with the calls for NA12891 for 8 epochs. The training samples are labeled with 10,824 DEL and 1,362 DUP calls in total. The true negatives are the rest of the exons for which no calls are made. We test the model on the MetaSV-based CNV calls provided for NA12892.

**Fine tuning and test sets from the samples in Guo et al. bladder cancer call set.** Guo et al. report matched WES and WGS samples of 16 bladder cancer patients (accession number: SRP017787). We fine-tune the ECOLE model with 3 cancer samples (samples B112, B24, B80) from this data set[31]. We use the semi-ground truth labels obtained on the matched WGS samples for these 3 patients for fine-tuning. The training samples are labeled with 23,383 DEL and 282,573 DUP calls in total. The true negatives are the rest of the exons for which no calls are made.

We use the remaining 13 cancer samples for testing in two ways. First, we check if tools make calls in the RT-qPCR-validated regions in these samples. Then, we use CNVnator to obtain the semi-ground truth labels on the matched WGS samples for these 13 patients to compute precision, recall and F1-scores. The test samples are labeled with 150,106 DEL and 974,332 DUP calls in total. See Supplementary Dataset 1 for the respective names of the samples.

### Experimental setup
**Compared methods.** We compared ECOLE with the following state-of-the-art WES-based germline CNV callers: XHMM v1.0, CODEX2, CONIFER v0.2.2, GATK v4[17,25,26,29]. These report categorical CNV predictions like ECOLE (i.e., deletion, duplication, or no-call). CNV-kit v0.9.7 and Control-FREEC v11.5[27,28] are also WES-based germline CNV callers but they report exact (i.e., integer) CNVs. To be able to also compare with these two tools, we discretize their predictions. That is, if the predicted copy number is larger than 2, it is classified as duplication; if it is less than 2, it is classified as deletion and no-call if it is equal to 2. We polished the calls made by all of the aforementioned tools using DECoNT as described in ref. 16 and used the DECoNT models released on GitHub. Polished call sets of these methods are called DECoNT-*toolname* (e.g., DECoNT-XHMM). We also compared ECOLE with CNLearn v1 which learns to aggregate the calls of other WES-based germline CNV callers (CANOES, XHMM, CONIFER and CLAMMS). Through personal communication, we obtained the calls of this tool on 28 samples in our test set.

**Parameter Settings.** For all samples we align, WES reads to the reference genome (GRCh38) using BWA with the -mem option and default parameters. We calculate the read depth using the Sambamba tool[38] with the base -L option to align the reads in the exon regions. We ran the compared methods using their recommended settings. For XHMM, the following parameter values were used: $Pr(\text{start DEL}) = Pr(\text{start DUP}) = 1e-08$; mean number of targets is 6; mean distance between targets is 70kb, and DEL, DIP, DUP read depth distributions were modeled as $\sim \mathcal{N}(-3,1)$, $\sim \mathcal{N}(0,1)$ and $\sim \mathcal{N}(3,1)$, respectively. For

CODEX2, the minimum read coverage was set to 20. CoNIFER performs SVD on the data to remove top $n$ singular vectors. We set $n$ to 6. For GATK, Control-FREEC and CNV-kit, we set all parameters to default values. We rely on the default parameter sets for the methods based on the comparisons done in Ozden et al., (2022) on the best-performing method, which shows that (i) the default parameter setting resulted in the best performance on the 1000 Genomes Dataset which we also use, and (ii) that the polished call sets results in better performance compared to the raw call set regardless of the parameter setting being relaxed or conservative.

**Training ECOLE.** We trained our model using the WES data as the training set of 550 samples from the 1000 Genomes data set. We used the Adam optimizer[39] and the model converged in 4 epochs. We used Xavier weight initialization[40]. We started training with a learning rate of $5 \cdot 10^{-5}$ and used a cosine annealing learning rate schedule.

To obtain the final ECOLE[FT-EXPERT] model, we further fine-tuned the ECOLE model with golden standard ground truth calls on 4 samples obtained from Chaisson et al. as explained in the Data Sets section. We again used Adam optimizer and cosine annealing schedule with an initial earning rate of $5 \cdot 10^{-5}$. The model converged in 11 epochs.

To obtain the final ECOLE[FT-SOMATIC] model, we further fine-tuned the ECOLE model with the semi-ground truth calls made on 3 cancer samples obtained from Guo et al. as explained in the Data Sets section. We have used the Adam optimizer and cosine annealing learning rate scheduler with a base learning rate of $5 \cdot 10^{-5}$, fine-tuning lasted for 11 epochs.

All models are trained on a SuperMicro SuperServer 4029GP-TRT with 2 Intel Xeon Gold 6140 Processors (2.3GHz, 24.75M cache) and 256GB RAM. We used a single NVIDIA GeForce RTX 2080 Ti GPU (24GB, 384Bit) for training. The initial model took approximately 15 days to converge and each fine-tuning took approximately 4 hours. Note that users do not need to train a model from scratch and can use the released ECOLE model for inference which is rapid. The average time to call all CNVs per exome is ~5 mins.

**Performance Metrics.** ECOLE assigns a pseudo probability score for each call to be deletion, duplication, or no-call where the event with the largest score is the final prediction. We measured the performance of all compared methods and ECOLE using precision and recall which are defined as follows:

$$\text{Duplication precision} \left( PRE_{dup} \right) = \frac{TP_{dup}}{TP_{dup} + FP_{dup}} \quad (1)$$

$$\text{Deletion precision} \left( PRE_{del} \right) = \frac{TP_{del}}{TP_{del} + FP_{del}} \quad (2)$$

$$\text{Overall precision} = \frac{PRE_{dup} + PRE_{del}}{2} \quad (3)$$

$$\text{Duplication recall} \left( REC_{dup} \right) = \frac{TP_{dup}}{T_{dup}} \quad (4)$$

$$\text{Deletion recall} \left( REC_{del} \right) = \frac{TP_{del}}{T_{del}} \quad (5)$$

$$\text{Overall recall} = \frac{REC_{dup} + REC_{del}}{2} \quad (6)$$

where $TP_{dup} :=$ the number of duplication calls that are correctly called; $TP_{del} :=$ the number of deletion calls that are correctly called; $FP_{dup} :=$ the number of duplication calls that are incorrectly called;

$FP_{del} :=$ the number of deletion calls that are incorrectly called; $T_{dup} :=$ the number of ground truth duplication calls; $T_{del} :=$ the number of ground truth deletion calls.

## ECOLE Architecture

**Problem Formulation.** Let $X$ be the set of all exons with available read depth signal and $X^i$ indicate the $i^{th}$ exon where $i \in \{1,2,....,N\}$ and $N = |X|$. Every $X^i$ is associated with the following features: $X^i_{chr}$, $X^i_{start}$, $X^i_{end}$ and $X^i_{RDSeq}$. $X^i_{chr}$ is the chromosome of the exon where $chr \in \{1, 2, 3,...,24\}$. 23 and 24 represent chromosomes X and Y, respectively. $X^i_{start}$, $X^i_{end}$ are the start and end coordinates of the exonic region. $X^i_{RDSeq}$ is a standardized vector of read depth values at a base pair resolution. Standardization is performed for every read depth value using the global mean and standard deviation of read depth values in the training data. Every $X^i_{RDSeq}$ is $-1$ padded from left to have the maximum length of 1000. We experiment with different maximum length values and proceed with 1000 as, overall, it performs best for ECOLE. Please see Section 3.5 of the Supplementary Text for these experiments. For exons longer than 1000 bps, they are considered if the non-zero read-depth values in that exon are of length < 1000. $Y^i$ represents the corresponding ground truth label for exon $i$, either obtained from CNVnator or from Chaisson et al. depending on the application. Let $\hat{Y}^i = f(X^i, \theta)$ be the CNV prediction (i.e deletion, duplication, no-call) using the model $f$ (a multi-class classifier) which is parameterized by $\theta$. The goal is to find the model parameters $\theta$ that minimize the difference between predicted exon-level CNV labels and their ground truth labels.

**Model Description.** We illustrate the overview of the model in Fig. 1. Each exon $i$ is associated with the vector $X^i_{RDSeq} \in \mathbb{R}^{1000 \times 1}$ which represents the read depth signal in that region.

First, ECOLE maps each read depth value $j$ of the read depth vector for the $i^{th}$ exon ($X^i_{RDSeq}[j]$) into a higher dimension $H = 192$ by using a fully connected neural network (see Eq. (7)).

$$FFN\left(X^i_{RDSeq}[j]\right) = \left(X^i_{RDSeq}[j] \cdot W + b^T\right)^T, \qquad W \in \mathbb{R}^{192 \times 1}, b \in \mathbb{R}^{192} j \in [1,1000] \quad (7)$$

We refer to the transformed form of the full vector $X^i_{RDSeq}$ as the input embedding and denote it with $\mathbf{X}^i_{embed} \in \mathbb{R}^{192 \times 1000}$ (see Eq. (8)).

$$\mathbf{X}^i_{embed} = FFN(X^i_{RDSeq}[1]) \ldots FFN(X^i_{RDSeq}[1000]) \quad (8)$$

ECOLE employs two techniques to learn the context in which the read depth signal indicates a copy number variation. First, it learns a *Chromosome Specific Classification Token* matrix $C \in \mathbb{R}^{192 \times 24}$ where each chromosome $k$ is represented with a column vector $c^k$ of size 192. That is, $c^k = C[:,k]$ where $c^k \in \mathbb{R}^{192}$. The vector for the chromosome in which exon $i$ resides ($c^{X^i_{chr}}$) is concatenated with $\mathbf{X}^i_{embed}$ to obtain $\hat{\mathbf{X}}^i_{embed} \in \mathbb{R}^{192 \times 1001}$ (see Eq. (9)). This joint matrix lets the model learn the meaning of the read depth vector in the context of different chromosomes to be able to distinguish chromosome-specific read depth patterns and model the variance across chromosomes.

$$\hat{\mathbf{X}}^i_{embed} = \mathbf{X}^i_{embed} c^{X^i_{chr}} \quad (9)$$

The second technique is using a positional encoding which enables the model to learn the relative locations of the read depth values with respect to each other and absolute position in the entire exome sequence and extract the position meaning that contributes to calling CNVs. In this work, for an exon $i$, we create a location vector $v$ of length 1001. We use sine and cosine functions of different radial frequencies similar to the version in[20] to create the positional embedding

matrix $\mathbf{E}_{pos}^i \in \mathbb{R}^{192 \times 1001}$ as done in Eqs. (10) and (11).

$$\mathbf{E}_{pos}^i[loc,j] = \sin(loc/10^{9*2j/H}), \qquad loc \in \{1,\ldots,1001\}, j \in \{1,\ldots,H/2\} \tag{10}$$

$$\mathbf{E}_{pos}^i[loc,2j+1] = \cos(loc/10^{9*2j/H}), \qquad loc \in \{1,\ldots,1001\}, j \in \{1,\ldots,H/2\} \tag{11}$$

To serve as an intuition, we could assume that positional encoding is a clock, then *loc* and *j* are hour and minute hands, respectively. Moving along the *loc* (i.e., over read depth embedding) and *j* (i.e. between 1 and latent dimension *H*) values is basically rotating the hour and minute hands with varying frequencies. The constant $10^9$ allows the encoding to uniquely map the start and end coordinates, $X_{start}^i$, $X_{end}^i$, which have a range of $[14.6 \cdot 10^3, 290 \cdot 10^6)$. This encoding enables the model to get positional and deal with the inherent noise in varying read depth values. This matrix is summed with $\hat{\mathbf{X}}_{embed}^i$ to obtain the input to the transformer $\mathbf{O}_0^i \in \mathbb{R}^{192 \times 1001}$ (see Eq. (12)).

$$\mathbf{O}_0^i = \hat{\mathbf{X}}_{embed}^i + \mathbf{E}_{pos}^i \tag{12}$$

ECOLE uses an efficient variant of the Transformer architecture[41] (only the encoder part). The encoder consists of a sequence of a parallel attention block (Multi-head attention, $MHA^{(h)}$) followed by a multi-layered perceptron (MLP) block. The multi-head attention mechanism lets the model learn the pertinence of read depth values in a chromosome in relation to deletion and duplication events (see Fig. 1 and[20] for MHA details). That is, it learns which parts of the signal it needs to focus on. An MHA block uses 8 parallel attention layers (i.e. heads). Firstly, the inputs for MHA are layer normalized (LN) and are propagated through the MHA block. Later, the outputs of these blocks are summed with the input of the respective LN block. The summed output is again layer normalized and then passed through an MLP. The outputs of the MLP block are summed with the input of the respective LN block to produce $\mathbf{O}_1^i \in \mathbb{R}^{192 \times 1001}$ (See Eqs. (13) and with the input of the respective LN block (14)). This procedure is repeated *L* times where $L = 3$ in our application.

$$\mathbf{O}_\ell^{i\prime} = MHA(LN(\mathbf{O}_{\ell-1}^i)) + \mathbf{O}_{\ell-1}^i, \qquad \ell = 1,\ldots,L \tag{13}$$

$$\mathbf{O}_\ell = MLP(LN(\mathbf{O}_\ell^{i\prime})) + \mathbf{O}_\ell^{i\prime}, \qquad \ell = 1,\ldots,L \tag{14}$$

ECOLE passes the column vector corresponding to the chromosome of exon $i$ ($\mathbf{O}_3[:, X_{chr}^i] \in \mathbb{R}^{192 \times 1}$) through an MLP to obtain probabilities for deletion, duplication and no-call events and maximum among these is returned as the prediction for that exon $i$ (see Eq. (15)).

$$\hat{Y}^i = \arg\max(Softmax(MLP(O_3^i[:, X_{chr}^i]))) \qquad \hat{Y}^i \in \{DEL, DUP, NOCALL\} \tag{15}$$

### Processing exons with no read depth available

We developed ECOLE to perform CNV calling on exon target regions using read depth information, however, it is important to note that about 20% of the exon targets do not contain read depth sequences on average per sample. In order to perform CNV calling to these regions, we have applied majority voting on the predictions of ECOLE based on the 3 nearest neighbor exon targets. Supplementary Fig. 25 demonstrates this procedure visually for further reference.

### Baseline

We compare ECOLE with various other models which can be considered as baselines to show the need for a deep-learning method and a complex architecture like transformers to call CNVs on WES data. We used the same training and test set we used for training and testing of ECOLE in all experiments.

First, we use a linear SVM classifier and XGBoost as baseline methods. Please see Section 3.2 of the Supplementary Text for the details of hyperparameter selection for baseline models. We input the read depth signal per exon and use the exon level semi-ground truth CNVnator labels for performance comparison on the 1000 Genomes Dataset test set as done in Section 2.2. The precision and recall performances are shown in Supplementary Table 29. We observe that SVM achieves 58.5% overall recall, and 3% precision. On the other hand, XGBoost method achieves relatively higher precision rate, which is 64%, yet it is only able to yield 1.9% recall. ECOLE has 62% recall and 77% precision which corresponds to a 63.3% and 65% F1-score improvements over SVM and XGBoost baseline methods, respectively. This shows the need for a complex model like ECOLE to learn an attention-based embedding on the read depth signal to be able to accurately classify exons as deleted or duplicated.

We also compare ECOLE's transformer-based architecture against convolutional neural network (CNN)-based solutions. First, we train a CNN model which only takes the read depth as input. We train two models separately (i) for chromosome 21 where ECOLE performs the best (88.4% F1-score), and (ii) for chromosome 10 where ECOLE performs the worst (6.67% F1-score). We observe that CNN-based models perform much worse than ECOLE in their respective chromosomes. Please see Supplementary Table 30 for detailed results and Supplementary Note 3.3 for the details about the CNN architecture.

We also train a CNN model which considers all chromosomes. This CNN model takes the read depth vector as input and the chromosome information is provided as a one-hot encoded vector which is concatenated with the read depth vector. The model achieves 5.9% precision and 6.5% recall (6.2% F1-score). We think that even though regions with a sharp read depth change is very important for CNV prediction, it's not enough. The context, that is the, read depth information of other exon regions is also important for successful prediction. The details about the architecture of the CNN model can be seen in Supplementary Notes 3.4.

### Ablation study and the need for a complex model like ECOLE

ECOLE uses a rather customized model of the transformer architecture which was first introduced in ref. [42] and proved its success in various domains from NLP to computer vision. Our transformer encoder uses (i) a chromosome-specific classification token, instead of a fixed classification token, and (ii) an exon-specific positional encoding instead of a fixed positional encoding to learn which parts of the signal are important for a CNV call and in which context (i.e., chromosome). To show the need for these context-specific techniques, we train a standard transformer architecture that does not have the aforementioned customized methods and which is otherwise identical to our model. That is, we use a standard 3-layered transformer model with fixed positional encoding. The model generates an output vector, which is the standard classification token (i.e., not chromosome-specific) of size $\in \mathbb{R}^{192}$. We start the training with the learning rate of $5 \cdot 10^{-5}$ and use a cosine annealing scheduler. We used the Adam optimizer during the training of the base model, and the model converged after 6 epochs.

Supplementary Table 31 shows the precision and recall performances of the baseline transformer model and ECOLE. We observe that ECOLE is able to outperform the baseline by a large margin, providing 30% average precision and 62% average recall improvements. The chromosome-specific classification token provides a good prior for the model to learn the relevance of each read depth value in the context of their respective chromosome. Moreover, exon-specific positional encoding renders the model to differentiate the absolute position of the exon samples along the exome. Hence, it gives the model the capacity to learn the context of the read-depth values along with the chromosome-specific classification tokens. As the read-depth samples can have variant distributions depending on the context (i.e.,

absolute position and the chromosome), the model is able to learn context-dependent sample distributions.

We also consider a version of the model where we provide the GC content of each exon to the model concatenated with the read depth vector. The rest of the model is kept the same. Here, we want to see if the model can use this extra information about the structure of the sequence to make better predictions. We obtained 52.2% DEL and 49.3% DUP F1-scores. As the performance does not improve compared to the original model, we prefer the model that does not use the GC content information.

## Interpretability of the ECOLE

In order to explain the predictions of our model, we use the Generic Attention-model Explainability method[33]. This method is class-dependent, uses label information, and it is a saliency method that highlights the relevant parts of the input for the classification that predicts the respective label. In addition, we use the Attention matrices **A** of the last Transformer blocks and obtain the specified relevance scores. The derivations for the relevancy scores for the efficient variant of the Transformer block, i.e. Performer[41] can be seen in Supplementary Note 3.1.

## Statistics & reproducibility

We used aligned WES reads from 1000 Genomes Project. We selected the first 1000 samples from the alphabetically ordered list of WGS samples (HG00096 to HG02356). We used 707 samples out of these 1000, for which matched WES samples were available. No data were excluded from the analyses. We randomly selected 550 WES samples for training and 157 for testing. The investigators were not blinded to allocation during experiments and outcome assessment.

## Reporting summary

Further information on research design is available in the Nature Portfolio Reporting Summary linked to this article.

## Data availability

All the data for reproduction is available under the license CC BY-NC-SA 2.0 at https://zenodo.org/record/8202814 (https://doi.org/10.5281/zenodo.8202814). Please note that ECOLE software is completely free for academic usage. However, it is licensed for commercial usage. Source data are provided with this paper for reproducing all Figures in the manuscript and Supplementary Info.

**Features:** The input features of the models is the WES read depth data which is generated using the Sambamba tool (default options).

The 1000 Genome Project sample names we used to train and test the models are provided in the Methods section which are available at the 1000 Genomes Project. WES and WGS samples are available at the following links: https://ftp.1000genomes.ebi.ac.uk/vol1/ftp/data_collections/1000_genomes_project/data/.

GiaB CNV call sets for the Ashkenazi family are available under accession codes SAMEA1573615 and SAMEA1573616 [https://ftp-trace.ncbi.nlm.nih.gov/giab/ftp/technical/svclassify_Manuscript/Supplementary_Information/metasv_trio_validation/]. Guo et al. samples are available at Sequence Reads Archive under accession code SRP017787 [https://www.ncbi.nlm.nih.gov/sra/?term=SRP017787].

**Labels:** The labels we use for training, fine-tuning, and testing are available at the following link: https://zenodo.org/record/8202814. The 1000 Genome Project labels, Guo et al., Chaisson et al. (under accession code Nstd152 [https://www.ncbi.nlm.nih.gov/dbvar/?term=nstd152]) and Ashkenazi family labels can be accessed on the Zenodo repository.

## Code availability

**Software:** ECOLE is implemented and released at https://github.com/ciceklab/ECOLE[43] under CC-BY-NC-ND 4.0 International license. The scripts used to generate the data for all figures and tables in the manuscript and the source code are provided at https://zenodo.org/record/8202814 https://doi.org/10.5281/zenodo.8202814.

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

## Acknowledgements

We would like to thank Vijay Kumar and Dr. Santhosh Girirajan for their help with obtaining results on CNLearn. We also thank Dr. Girirajan for the feedback on our manuscript. We thank Arcin U. Erguzen for his amazing design skills and efforts in preparing the system figure.

## Author contributions

A.E.C. and C.A. supervised the study. B.M. and F.O. designed the model. B.M., F.O., G.K. and M.A.Y. implemented the software and performed the experiments. A.E.C., C.A., B.M., F.O., G.K. and M.A.Y. wrote the manuscript.

## Competing interests

Furkan Ozden, Can Alkan and Ercument Cicek are cofounders of Lidya Genomics. The authors declare no competing interests.
