## [Peer Review File · Nature Communications]

ECOLE: Learning to call copy number variants on whole exome sequencing dataEditorial Note: This manuscript has been previously reviewed at another journal that is not operating a transparent peer review scheme. This document only contains reviewer comments and rebuttal letters for versions considered at *Nature Communications*.

Reviewer #1 (Remarks to the Author):

The responses from Mandiracioglu et al. concerning their new deep learning method, ECOLE, for calling CNVs at a per-exon (ROI) level in WES data, have addressed all of my concerns from their initial submission. Additionally, in response to reviewers queries, the authors have made essential corrections, enriched the text with further details, and appended new materials and supplementary data to their manuscript. This revised version of the manuscript is considerably stronger and more informative, offering deeper insights into the performance of ECOLE in comparison to other methods. These include the recent GATK CNV caller and the group's previous CNV-call polishing method, DECONT.

The results demonstrate that ECOLE outperforms other previously published CNV callers for WES data. Notably, through transfer learning, the method significantly improves its accuracy using either manually curated or orthogonally validated calls. This flexibility allows ECOLE to tackle somatic CNV calling and potentially specialized gene-panel calls derived from WES data. The newly incorporated analysis involving the Genome-in-a-Bottle Ashkenazim trio is highly informative and underscores the value of transfer learning on a DL method which was initially trained on a "silver" standard.

The authors have also conducted a comprehensive evaluation of the method's advantages and limitations. This includes assessing performance across different types (deletion or duplication) and size ranges of CNVs. As expected, ECOLE, like all CNV callers, faces challenges in identifying very short, single-exon events due to limited window information and the absence of a sufficient ground truth for training in this size range.

Regarding my previous suggestion to incorporate SV data from the SEQQC2 consortium, I note that this data has now been published (<https://genomebiology.biomedcentral.com/articles/10.1186/s13059-022-02816-6>). However, upon my own inspection, it seems the consortium has only made curated somatic calls for the cancer cell line available. Paradoxically, they have not provided consensus SV calls for the corresponding germline cell line. This limits the utility of this data for benchmarking SV callers. As such, I will not insist on including this data in the current ECOLE manuscript.

Lastly, the authors' response to my question about the varying performance of CNV calling across different human chromosomes—potentially addressable by chromosome-based training—has been satisfactory. Although this approach did not improve the situation, the issue remains puzzling. To my knowledge, no one has previously conducted a chromosome-by-chromosome analysis of CNV calling performance. I believe it will be valuable for the broader research community to become aware of this phenomenon through this publication.

Reviewer #2 (Remarks to the Author):

The authors have address all of my concerns.

Reviewer #3 (Remarks to the Author):

Mandiracioglu et al. have addressed most of the comments. The authors have toned down their claims of clinical applicability of the model. Despite the model probably not be suitable for all clinical applications the improvement over the state of the art is substantial and definitively a novel and relevant finding.

Could the authors please clarify the following points:

Re comment 2 a)

It is reassuring to see that the overall performance is not largely different between different WES capture methods. Nevertheless, a more fine grained analysis of the impact of the design on the

model performance would be informative for future improvements of the model. I would like to see performance evaluation in different classes of exons separately. For example: exons that are covered by a single capture probeset vs exons that are covered by multiple probesets. Please also check if probesets are overlapping with each other or not. And please also stratify exons by their GC content or predicted free energy of the double helix (based on sequence).

Re comment 2 c)

I would like to thank the authors for providing this additional analysis. It is unclear how the GC content was computed from fastq (WES reads). The GC content should be computed from the reference genome sequence and the exon annotation. Please clarify if this is the case.

Re comment 3)

It seems that ECOLE really only works if fine tuning data is available. I think this is a (small) limitation, as it cannot simply be applied to new data, without fine tuning. Does this imply that a lab that wishes to perform WES CNV calling needs to first generate data and high quality annotations to be able to use ECOLE successfully? Please add this to the discussion and explain how labs can profit from the tool in practice.

Re minor comment 3 and 4)

It looks like the majority of exons is actually in the size bin 100-200bp. The DEL F1 score there is not as high as in other bins. This could potentially be improved by using smaller fragment lengths. Please assess this for a few exemplary fragment sizes below 1000.

New comment:

In the discussion: "While it is possible to change a false positive to a true positive call, it is rare and it is not possible to change a false negative call (i.e., no call) to a true positive call."

This is confusing. If a call was a false positive, it means that the true label is negative. So it cannot be changed to a true positive but only to a true negative because only the prediction and not the true label can be changed.

Reviewer Response Document for

ECOLE: Learning to call copy number variants on whole exome sequencing data

We would like to thank the reviewers for their constructive feedback. Please find our responses below.

Reviewer #1 (Remarks to the Author):

The responses from Mandiracioglu et al. concerning their new deep learning method, ECOLE, for calling CNVs at a per-exon (ROI) level in WES data, have addressed all of my concerns from their initial submission. Additionally, in response to reviewers queries, the authors have made essential corrections, enriched the text with further details, and appended new materials and supplementary data to their manuscript. This revised version of the manuscript is considerably stronger and more informative, offering deeper insights into the performance of ECOLE in comparison to other methods. These include the recent GATK CNV caller and the group's previous CNV-call polishing method, DECONT.

The results demonstrate that ECOLE outperforms other previously published CNV callers for WES data. Notably, through transfer learning, the method significantly improves its accuracy using either manually curated or orthogonally validated calls. This flexibility allows ECOLE to tackle somatic CNV calling and potentially specialized gene-panel calls derived from WES data. The newly incorporated analysis involving the Genome-in-a-Bottle Ashkenazim trio is highly informative and underscores the value of transfer learning on a DL method which was initially trained on a "silver" standard.

The authors have also conducted a comprehensive evaluation of the method's advantages and limitations. This includes assessing performance across different types (deletion or duplication) and size ranges of CNVs. As expected, ECOLE, like all CNV callers, faces challenges in identifying very short, single-exon events due to limited window information and the absence of a sufficient ground truth for training in this size range.

Regarding my previous suggestion to incorporate SV data from the SEQQC2 consortium, I note that this data has now been published (<https://genomebiology.biomedcentral.com/articles/10.1186/s13059-022-02816-6>). However, upon my own inspection, it seems the consortium has only made curated somatic calls for the cancer cell line available. Paradoxically, they have not provided consensus SV calls for the corresponding germline cell line. This limits the utility of this data for benchmarking SV callers. As such, I will not insist on including this data in the current ECOLE manuscript.

Lastly, the authors' response to my question about the varying performance of CNV calling across different human chromosomes—potentially addressable by chromosome-based training—has been satisfactory. Although this approach did not improve the situation, the issue remains puzzling. To my knowledge, no one has previously conducted a chromosome-by-chromosome analysis of CNV calling performance. I believe it will be valuable

for the broader research community to become aware of this phenomenon through this publication.

Thank you.

Reviewer #2 (Remarks to the Author):

The authors have addressed all of my concerns.

Thank you.

Reviewer #3 (Remarks to the Author):

Mandiracioglu et al. have addressed most of the comments. The authors have toned down their claims of clinical applicability of the model. Despite the model probably not be suitable for all clinical applications the improvement over the state of the art is substantial and definitively a novel and relevant finding.

Could the authors please clarify the following points:

Re comment 2 a)

It is reassuring to see that the overall performance is not largely different between different WES capture methods. Nevertheless, a more fine grained analysis of the impact of the design on the model performance would be informative for future improvements of the model. I would like to see performance evaluation in different classes of exons separately. For example: exons that are covered by a single capture probeset vs exons that are covered by multiple probesets. Please also check if probesets are overlapping with each other or not. And please also stratify exons by their GC content or predicted free energy of the double helix (based on sequence).

We analyzed the impact of the number of probes within exons on the performance of ECOLE using again the NA12878 sample sequenced with (i) NimbleGen SeqCap v3 and (ii) SeqCap EZ Human Exome Library v3.0 capture kits. We now discuss these new findings in Section 2.2 of the manuscript.

We observed that SeqCap EZ Human Exome Library v3.0 and NimbleGen SeqCap v3 capture kits cover 67.8% and 99.3% of the exome by single probes, respectively. About 85% of probesets are overlapping with each other.

As can be seen in the following table (now Supplementary Table 12), probe count does not substantially affect the model performance for the SeqCap EZ Human Exome Library v3 capture kit. For NimbleGen SeqCap v3, performance on single probe-covered exons is better compared to multiple probe-covered exons. There are only a few multiple probe-covered exons (0.7%) by NimbleGen SeqCap v3, and mostly reside in chromosomes 9 and 10 (51%). As discussed in Section 2.5 and shown in Figure 5, ECOLE performs relatively worse even when predicting

samples sequenced with the same capture kit. This might explain the low performance in multiple probe-covered exons in NimbleGen SeqCap v3.

Supplementary Table 12. The performance comparison of ECOLE on exons sequenced with single and multiple probes using SeqCap EZ Human Exome Library v3.0 and NimbleGen SeqCap v3 capture kits.

Performance Metrics	Capture Kits			
	SeqCap EZ Human Exome Library v3.0		NimbleGen SeqCap v3	
	Single Probe Exons	Multiple Probe Exons	Single Probe Exons	Multiple Probe Exons
DEL Precision	0.875	0.845	0.927	0.500
DUP Precision	0.541	0.559	0.513	0.636
Overall Precision	0.708	0.702	0.720	0.568
DEL Recall	0.530	0.484	0.554	0.024
DUP Recall	0.588	0.524	0.538	0.135
Overall Recall	0.559	0.504	0.546	0.080
DEL F1 Score	0.660	0.616	0.694	0.045
DUP F1 Score	0.563	0.541	0.526	0.222
Overall F1 Score	0.612	0.579	0.610	0.134

We also stratified exons with respect to their GC contents as suggested and compared ECOLE's performance on varying GC content rates for both capture kits. We observed that GC content does not substantially affect the overall performance. The results can be seen in the following figures (now Supplementary Figures 6 and 7 in the manuscript).

Supplementary Figure 6. The performance of ECOLE on exons with varying GC content rates for NA12878 sample sequenced with NimbleGen SeqCap v3 capture kit.

Supplementary Figure 7. The performance of ECOLE on exons with varying GC content rates for NA12878 sample sequenced with SeqCap EZ Human Exome Library v3.0 capture kit.

Re comment 2 c)

I would like to thank the authors for providing this additional analysis. It is unclear how the GC content was computed from fastq (WES reads). The GC content should be computed from the reference genome sequence and the exon annotation. Please clarify if this is the case.

You are right. We are sorry for the mistake in the previous review responses document. Indeed, we calculated GC content on the reference genome HG38.

Re comment 3)

It seems that ECOLE really only works if fine tuning data is available. I think this is a (small) limitation, as it cannot simply be applied to new data, without fine tuning. Does this imply that a lab that wishes to perform WES CNV calling needs to first generate data and high quality annotations to be able to use ECOLE successfully?

This is actually not fully correct. It depends on what we mean by “data”. ECOLE model would work on any WES “data” produced by any lab with no fine-tuning and would mimic the decision-making strategy of the CNVnator algorithm (for germline CNV calling on WES data).

If we mean “a new decision-making strategy” by “data” then we need fine-tuning. For instance, if a lab thinks that CNVnator labels/annotations for CNVs are incorrect and wants to perform manual labeling by eyeing the read depth over regions using IGV, then they’d need to fine-tune the base ECOLE model with a few annotated samples to tweak the decision making towards their strategy.

Please add this to the discussion and explain how labs can profit from the tool in practice.

We added the following paragraph to the Discussion Section.

“If one would like to use an ECOLE model with a different CNV calling strategy than available ECOLE models, they can label CNVs on a few WES data with their technique and fine-tune the base ECOLE model. This would tweak the decision-making of ECOLE towards their strategy. The tool is easily configurable to other specific CNV calling techniques without generating WES data.”

Re minor comment 3 and 4)

It looks like the majority of exons is actually in the size bin 100-200bp. The DEL F1 score there is not as high as in other bins. This could potentially be improved by using smaller fragment lengths. Please assess this for a few exemplary fragment sizes below 1000.

Thank you for the suggestion. We now added a new subsection to Supplementary Text to discuss this: Supplementary Text 3.5 - Performance Comparison of ECOLE models trained with

varying-sized input vectors. We also added the table below to the Supplementary Material as Supplementary Table 33, which provides detailed performance results.

We refer to this new section in Section 4.3 of the manuscript as follows:

“Every X_{RDSeq}^i is -1 padded from left to have the maximum length of 1000. We experiment with varying maximum length values and proceed with 1000 as, overall, it performs best for ECOLE. Please see Section 3.5 of the Supplementary Text for these experiments.”

Supplementary Text Section 3.5 reads as follows:

“We trained ECOLE models that input 100-, 200-, 500-, and 1000-sized input read depth vectors. We use the same hyperparameters discussed in Section 4.2 of the manuscript. We then test these models using our test samples.

We observe that ECOLE trained with a 100-sized read depth vector performed best for calls smaller than 100bp with an overall F1-score of 31.4%. In contrast, ECOLE trained with the 1000-sized read depth vector achieved a 12.6% overall F1 score for these calls. We would like to note that the number of ground truth CNV calls for this region (<100bp) is very small (3253) compared to regions sized between 100-200 bp (970571).

For calls that are between 100 bp and 200 bp in size, ECOLE trained with a 1000-sized read depth vector outperforms ECOLE models trained with 200-, and 500-sized read depth vectors with an overall F1-score of 61.5% (7.5% and 14.7% improvement, respectively). Note that the model trained with a 100-sized read depth vector is not applicable for this and larger size ranges.

Finally, for calls longer than 200 bp, ECOLE trained with the 1000-sized read depth vector performs better than the version trained with a 500-sized read depth vector with an overall F1 score of 63.9% (15.6% improvement).

While for very short exons, one can opt for the ECOLE model, which uses a 100-sized read depth vector, that model cannot make predictions for larger exons which is very restrictive. Since a 1000bp input vector covers most of the exons in the exome, and shows overall good performance, we provide results with this ECOLE version. Please see the table below for the deletion and duplication prediction performances of these models.”

Supplementary Table 33. The performance comparison of ECOLE models trained with varying-sized read depth vector inputs (100, 200, 500, and 1000). The performance is shown for calls shorter than 100bp, calls between 100bp and 200bp, and calls larger than 200bp.

Call sizes	ECOLE input size	DEL Precision	DEL Recall	DUP Precision	DUP Recall	Overall Precision	Overall Recall	DEL F1 Score	DUP F1 Score	Overall F1 Score
For calls <100	100	0.786	0.525	0.0	0.0	0.393	0.262	0.629	0.0	0.314
	200	0.987	0.268	0.0	0.0	0.493	0.134	0.421	0.0	0.210
	500	0.994	0.212	0.0	0.0	0.492	0.106	0.349	0.0	0.176
	1000	0.533	0.166	0.0	0.0	0.266	0.083	0.253	0.0	0.126
For calls 100-200	100	0.0	0.0	0.0	0.0	0.0	0.0	0.0	0.0	0.0
	200	0.783	0.486	0.755	0.352	0.769	0.419	0.599	0.480	0.540
	500	0.846	0.298	0.724	0.378	0.785	0.338	0.440	0.496	0.468
	1000	0.533	0.704	0.590	0.660	0.562	0.682	0.606	0.623	0.615
For calls >200	100	0.0	0.0	0.0	0.0	0.0	0.0	0.0	0.0	0.0
	200	0.0	0.0	0.0	0.0	0.0	0.0	0.0	0.0	0.0
	500	0.865	0.349	0.710	0.352	0.794	0.363	0.497	0.470	0.483
	1000	0.620	0.788	0.535	0.641	0.578	0.715	0.694	0.583	0.639

New comment:

In the discussion: "While it is possible to change a false positive to a true positive call, it is rare and it is not possible to change a false negative call (i.e., no call) to a true positive call."

This is confusing. If a call was a false positive, it means that the true label is negative. So it cannot be changed to a true positive but only to a true negative because only the prediction and not the true label can be changed.

We thank the reviewer for pointing out this mistake. We now corrected this sentence as "While it is possible to change a false positive to a true negative call, it is rare, and it is not possible to change a false negative call (i.e., no call) to a true positive call."

Reviewer #3 (Remarks to the Author):

I would like to thank the authors for addressing all of my comments.

**Reviewer Response Document for
ECOLE: Learning to call copy number variants on whole exome sequencing data**

We would like to thank the reviewers for their constructive feedback.

Reviewer #3 (Remarks to the Author):

I would like to thank the authors for addressing all of my comments.

Thank you.